# Histological, immunohistochemical and transcriptomic characterization of human tracheoesophageal fistulas

**Erwin Brosens**[1]*, **Janine F. Felix**[2,3], **Anne Boerema-de Munck**[2,4], **Elisabeth M. de Jong**[1,2], **Elisabeth M. Lodder**[1,5], **Sigrid Swagemakers**[6,7], **Marjon Buscop-van Kempen**[2,4], **Ronald R. de Krijger**[6,8], **Rene M. H. Wijnen**[2], **Wilfred F. J. van IJcken**[4], **Peter van der Spek**[6,7], **Annelies de Klein**[1], **Dick Tibboel**[2], **Robbert J. Rottier**[2,4]

1 Department of Clinical Genetics, Erasmus University Medical Center–Sophia Children's Hospital, Rotterdam, The Netherlands, 2 Department of Pediatric Surgery, Erasmus University Medical Center–Sophia Children's Hospital, Rotterdam, The Netherlands, 3 Department of Pediatrics and Generation R Study Group, Erasmus University Medical Center–Sophia Children's Hospital, Rotterdam, The Netherlands, 4 Department of Cell Biology, Erasmus University Medical Center Rotterdam, Rotterdam, The Netherlands, 5 Department of Clinical and Experimental Cardiology, Heart Center, Academic Medical Center, Amsterdam, The Netherlands, 6 Department of Pathology, Erasmus University Medical Center Rotterdam, Rotterdam, The Netherlands, 7 Department of Clinical Bioinformatics, Erasmus University Medical Center Rotterdam, Rotterdam, The Netherlands, 8 Dept. of Pathology, University Medical Center Utrecht, Utrecht, The Netherlands

* e.brosens@erasmusmc.nl

**Data Availability Statement:** Raw data is uploaded to the Gene Expression Omnibus (https://www.

## Abstract

Esophageal atresia (EA) and tracheoesophageal fistula (TEF) are relatively frequently occurring foregut malformations. EA/TEF is thought to have a strong genetic component. Not much is known regarding the biological processes disturbed or which cell type is affected in patients. This hampers the detection of the responsible culprits (genetic or environmental) for the origin of these congenital anatomical malformations. Therefore, we examined gene expression patterns in the TEF and compared them to the patterns in esophageal, tracheal and lung control samples. We studied tissue organization and key proteins using immunohistochemistry. There were clear differences between TEF and control samples. Based on the number of differentially expressed genes as well as histological characteristics, TEFs were most similar to normal esophagus. The BMP-signaling pathway, actin cytoskeleton and extracellular matrix pathways are downregulated in TEF. Genes involved in smooth muscle contraction are overexpressed in TEF compared to esophagus as well as trachea. These enriched pathways indicate myofibroblast activated fibrosis. TEF represents a specific tissue type with large contributions of intestinal smooth muscle cells and neurons. All major cell types present in esophagus are present—albeit often structurally disorganized—in TEF, indicating that its etiology should not be sought in cell fate specification.

ncbi.nlm.nih.gov/geo/) with accession number
GSE148247.

**Funding:** This work was supported by the Edgar
Doncker Foundation [DT; J.F.F.] and the Sophia
Foundation for Scientific Research [Grant Numbers
493 [DT; ADK; E.M.de J.], 436 [A.de M.] and S13-
09 [DT; ADK; E.B.] The funders had no role in study
design, data collection and analysis, decision to
publish, or preparation of the manuscript.

**Competing interests:** None of the authors have
financial, professional, or personal conflicts of
interest, all authors reviewed and approved the final
manuscript.

## Introduction

Esophageal atresia (EA) and tracheoesophageal fistula (TEF) are frequently occurring foregut malformations with an incidence of around 1 in 3,500 births [1–3]. On morphological grounds, five types of esophageal atresia are recognized, of which proximal atresia with a distal TEF is present in 85% of patients [4]. The atresia and TEF are surgically treated in the first days after birth. EA/TEF etiology is likely multifactorial with a strong genetic component [5, 6] and can be either an isolated congenital anatomical malformation or one of the component features of a (suspected) syndrome [7, 8]. Environmental factors have been suggested to play a role in the etiology of EA/TEF, although no single external factor has consistently been identified [3, 9–23]. The genetic etiology of isolated EA/TEF is largely unknown. Approximately 10% of patients with syndromal EA/TEF have chromosomal anomalies, mostly trisomies [1, 24, 25], deleterious Copy Number Variations (CNVs) [26–28] or a monogenetic syndrome [29–40]. Animal models support a genetic contribution [41–63], although, there is little overlap between genes implicated by animal models and the genes known to be involved in human disease [64]. Identification of genetic factors in patients is hampered by the large genetic and phenotypic heterogeneity, insufficient knowledge of disturbed biological processes, gene networks and initial cell type(s) affected. Even the exact mechanisms of normal development of the human foregut and its role in the etiology of EA/TEF are subject of discussion in the literature [65–72].

Up to now, there are only few molecular studies involving human TEF material, which often included small numbers of patients [73–75], and the results are contradictory. Histological studies of TEF and distal esophagus show a mixed contribution of different cell types. Human samples have been described to have (pseudo-) stratified squamous epithelium, tracheobronchial remnants, abnormal mucous glands, a disorganized muscular coat and cartilage [73, 76, 77], but ciliated epithelium has also been observed [77]. Using an unbiased approach, complemented with immunohistochemistry and RT-PCR, the expression of specific proteins and genes in the TEF has been studied in both animals and humans. These include NK2 Homeobox 1 (NKX2-1), Sonic HedgeHog (SHH) and members of the Bone Morphogenic protein (BMP) pathway [73–75, 78–80]. Most of these experiments seem to support a respiratory origin of the TEF in humans, but the number of human TEFs examined was small, ranging from one to nine. In a relatively recent study Smigiel et al. studied the expression pattern of 26 esophageal lower pouches and found enrichment of differentially expressed genes in Wingless and Int- (WNT), SHH and cytokine/chemokine signaling pathways [81].

To gain more insight in the origin of the TEF, we aimed to examine and describe TEF composition using a combination of whole-genome transcription profiling and (immuno-) histochemistry (see S1 Graphical abstract). We hypothesized that such characterization of human TEFs provides insight in the molecular and mechanistic etiology of EA/TEF.

## Materials and methods

### Human patient control sample characteristics

The protocol for this study was approved by the Medical Ethics Committee of the Erasmus MC Rotterdam, the Netherlands and the Dr. Behcet Uz Children's Hospital in Izmir, Turkey. Written (parental) consent was obtained. This study has been approved by the Erasmus University Medical Center's local ethics board (protocol no.193.948/2000/159, addendum Nos. 1 and 2.) After parental informed consent was obtained, tissue samples of the TEF of children with EA with a distal TEF were taken during primary operative repair of the EA/TEF. The operating surgeons, who have not been involved in the study, determined the safety and technical feasibility of removing the tissue.

We obtained 21 surgically resected tissue samples that were qualitatively suitable for transcriptome profiling. Patient characteristics including sex, gestational age, birthweight and associated anomalies are described in Table 1. All patients were term, except for 1, who was born at 32 weeks. The time of repair was between 2 and 16 days after birth with a median of 2 days. Previously, deleterious variation in the disease genes (*SOX2*, *CHD7*, *MID1*, *SALL1*, *MYCN*, *EFTUD2*, and the *FANC* genes) candidate genes has been determined in patients of which sufficient DNA material was available. Variants have either been determined with a Molecular Inversion Probe (MIP) gene panel [82], Sanger sequencing or Whole Exome Sequencing (WES) during routine diagnostic procedures. We did not have sufficient DNA of patient 3, 23 and 28. Moreover the Copy number profiles of all patients except patient 3 and 18 were determined previously [26]. Several patients had a rare CNV of uncertain significance. Patient SKZ_0106 was diagnosed with CHARGE syndrome. No other causal mutations were identified in these patients. Furthermore, three tracheal, three esophageal and four lung samples were selected to serve as control tissues. Control tissue (lung, trachea and esophagus) was received from the tissue bank of the Erasmus MC. Control samples were taken from autopsies of children of 17–25 weeks gestational age who had died of causes not related to trachea, esophagus or lung abnormalities and in whom there was no reason to assume any abnormalities of these organs.

**Table 1. Patient characteristics.**

| Patient | Gender | GA (wk+d) | BW (g) | IUGR | TSD | *Outcome* | *Associated anomalies* | | | | | | *Genetic anomalies* |
|---|---|---|---|---|---|---|---|---|---|---|---|---|---|
| | | | | | | | V | A | C | R | L | Other | |
| **SKZ_0399** | M | 38+4 | 3825 | - | 1 | Alive | - | + | - | - | + | A, B, D, E | Gain chr12:74018363–74108097 hg19 |
| **SKZ_0401** | M | 34+1 | 2060 | - | 3 | Alive | - | - | + | - | - | - | |
| **SKZ_1032** | M | 37+4 | 2640 | - | 2 | Alive | - | - | + | - | - | - | |
| **SKZ_0106** | F | 34+6 | 1200 | + | 3 | Deceased | - | - | - | - | - | | CHARGE syndrome |
| **SKZ_0150** | F | 38+0 | 2800 | - | 2 | Alive | - | - | + | - | + | - | |
| **SKZ_0416** | F | 33+5 | 1750 | - | 1 | Alive | - | - | - | + | - | B | Gain chr8:66955527–66980813 hg19 (*de novo*) |
| **SKZ_0286** | M | 35+6 | 1780 | + | 2 | Alive | - | - | + | + | - | A | |
| **SKZ_1344** | M | 42+0 | 3810 | - | 1 | Alive | - | - | - | - | - | A, B | |
| **SKZ_1003** | M | 37+2 | 3375 | - | 2 | Alive | + | - | - | - | - | C | Loss chr14:38928454–39044917 hg19 |
| **SKZ_1470** | M | 31+2 | 1780 | - | 1 | Alive | - | - | - | - | - | - | |
| **SKZ_1150** | F | 36+2 | 2120 | - | 2 | Alive | - | - | + | - | - | - | |
| **SKZ_0845** | F | 41+5 | 3170 | - | 2 | Alive | + | - | - | - | - | D | Loss chr12:74018363–74108097 hg19 |
| **SKZ_0123** | M | 37+1 | 2865 | - | 2 | Alive | - | - | - | + | - | - | Loss chr3: 8,975,742–9,024,521 h19 |
| | | | | | | | | | | | | | Gain chr16:56,937,855–57,151,796 hg19 |
| **SKZ_1248** | F | 37+5 | 2235 | + | 1 | Alive | - | - | - | - | - | - | Gain chr1:238656294–238780616 hg19 |
| | | | | | | | | | | | | | Loss chr10:19498889–20047506 hg19 |
| **SKZ_0703** | M | 42+3 | 3800 | - | 1 | Deceased | + | - | + | - | - | - | Gain chr3:1813064–2150011 hg19 |
| **SKZ_0673** | M | 40+2 | 3595 | - | 0 | Alive | + | - | - | - | - | A, B | |
| **SKZ_1466** | F | 41+0 | 3775 | - | 1 | Alive | - | - | - | - | - | - | |
| **SKZ_0546** | F | 40+5 | 3570 | - | 1 | Alive | - | - | - | + | - | - | |
| **SKZ_1037** | M | 40+4 | 3180 | - | 1 | Alive | + | - | - | + | - | - | |
| **SKZ_0720** | M | 40+1 | 3615 | - | 2 | Alive | - | - | - | - | - | - | |
| **SKZ_0876** | M | 36+1 | 1800 | + | 1 | Deceased | - | + | + | + | - | A, F | |

V: Vertebral/Rib; A: Anal; C: Cardiac; R: Renal; L: upper Limb; TSD: Time to surgery in days; F: female; M: male; GA: gestational age; wk: weeks; d: days; BW: birth weight; g: grams; IUGR: intra-uterine growth retardation; Time to surgery: time between birth and surgery; A: single umbilical artery; B: dysmorphic features (mild in patient no.1, 6 and 8); C: cleft lip, jaw and palate; D: toe anomalies; E: hypospadias; F: duodenal atresia; CHARGE: Coloboma, Heart defects, Atresia of choanae, Retardation, Genital anomalies, Ear anomalies.

In addition to these 21 patient samples, TEF material of 8 patients was available for histological staining. Control samples were paraffin-embedded samples of normal esophagus and trachea from autopsies of children born at term, who had died of unrelated causes. Also, control samples from preterm trachea and esophagus (gestational ages: 19 weeks+6 days and 17 weeks+3 days) were included, from individuals with normal overall development and without thoracic congenital anomalies. This material was obtained from the pathology department of the Erasmus MC in Rotterdam, the Netherlands.

## Transcriptome profiling

**RNA isolation and quality control.** All samples were snap frozen in liquid nitrogen and stored at –80˚C until further processing. Patient and control samples were homogenized on ice in TRIzol reagent (Invitrogen life technologies, Carlsbad, CA, USA) and total RNA was isolated following the manufacturer's instructions, but the organic extraction was repeated by adding 200μl of 0.1% DEPC water to increase RNA yield. RNA was purified using the Rneasy MinElute Cleanup kit (Qiagen, Valencia, CA, USA) and stored at –80˚C until further processing. RNA concentrations and OD 260/280 nm ratios were measured using the NanoDrop® ND-1000 UV-VIS spectrophotometer (NanoDrop Technologies, Wilmington, USA). RNA quality was evaluated by inspecting ribosomal 28S and 18S peaks and using the bioanalyzer (RNA integrity number (RIN) values above 8.0) Samples with low RNA quality were excluded from the transcriptome study.

Depending on the availability and/or quality of purified total RNA, cDNA was synthesized from 0.8–15 μg RNA using the GeneChip Expression 3'-Amplification.

Reagents One-Cycle cDNA Synthesis kit (Affymetrix, Santa Clara, CA, USA). Biotin-labelled cRNA synthesis, purification and fragmentation were performed according to standard protocols. Fragmented biotinylated cRNA was subsequently hybridized onto Affymetrix Human Genome U133 Plus 2.0 microarray chips, which were scanned with the Affymetrix GeneChip Scanner.

**Data processing and normalization**. The measured intensity values were analyzed using GeneChip Operating Software (GCOS). The percentage of present calls, background, and ratio of actin and GAPDH 3' to 5' indicated a high quality of samples and an overall comparability. Probe sets that were not present (according to Affymetrix MAS5.0 software) in any of the Genechips were omitted from further analysis. Raw intensities of the remaining probe set of each chip were log2 transformed and raw expression values were quantile normalized and transformed back to normal intensity values. Data analysis was carried out using BRB-array tools version 4.6.0 (October 2018) in combination with R version 3.5.1 (July 2018). For each probe set, the geometric mean of the hybridization intensities of all samples was calculated. The level of expression of each probe set was determined relative to this geometric mean and logarithmically transformed (on a base 2 scale) to ascribe equal weight to gene-expression levels with similar relative distances to the geometric mean. Raw data is uploaded to the Gene Expression Omnibus (https://www.ncbi.nlm.nih.gov/geo/; GSE148247. We used the algorithm embedded in the Ingenuity Pathway Analysis tool to infer enriched pathways Analysis settings and thresholds are provided in the supplementary methods.

**Class comparison of tissues types**. Genes whose expression differed by at least 1.5-fold from the median in at least 7% of the arrays were included in the analysis. Differential gene expression was determined among the classes (1) Esophagus, (2) TEF, (3) Lung, (4) Trachea and (5) all control tissues combined using a random-variance t-test (RV t-test). Genes were considered statistically significant if their p value was less than 0.05. Additionally, a global test using a p-value of 0.05 for each permutation (n = 10000) was used to confirm of whether the

expression profiles differed between the classes by permuting the labels of which arrays corresponded to which classes. Genes passing the individual tissue type comparison random-variance t-tests used to determine the number of differentially expressed genes between each class.

**Hematoxylin and eosin staining.** Samples were fixed in 10% buffered formaldehyde for two hours and after routine procedures embedded in paraffin. All paraffin blocks were cut into 4 μm sections.

Sections were deparaffinized in xylene and rehydrated. Routine hematoxylin and eosin staining were done and the sections were evaluated for different cell types and general structure of the tissue. Due to a limited amount of material, not all staining could be done on all samples.

### Immunohistochemistry

Sections were deparaffinized in xylene and rehydrated in ethanol. Antibody details are shown in S10 File. Endogenous peroxidase was blocked by 3% $H_2O_2$ in PBS for 20 minutes. For all proteins except SOX2, antigen retrieval was performed by heat induced epitope retrieval in a Tris/EDTA buffer (pH 9.0) for 20 minutes. Antigen-antibody complexes were visualized by a peroxidase-conjugated polymer DAB detection system (ChemMate DAKO Envision detection kit, Peroxidase/DAB, Rabbit/Mouse; Dako, Glostrup, Denmark). Immunohistochemistry for SOX2 was carried out using the Envision+ System (Dako, Glostrup, Denmark) and HRP-DAB colorimetric detection. Antigen unmasking was performed with microwave treatment in 10 mM citric acid buffer (pH adjusted to 6.0, 15 min at 600 W). Staining intensity was classified as negative and positive. Known positive tissues were used as controls. Due to a limited amount of material, not all staining could be done on all samples.

## Results

### Transcriptome analysis: Whole transcriptome comparison

We compared the transcriptomes of 21 TEFs to 3 esophageal, 3 tracheal and 4 lung samples. Unsupervised hierarchical clustering of all samples showed that TEF differ from the control samples (lung, trachea and esophagus) based on their whole genome transcription profiles (S1A in S1 File). All TEFs clearly clustered together separately from lung and tracheal tissue. Whereas the lung tissue samples also clustered separately, esophageal and tracheal tissue showed more mixed patterns. We compared the expression patterns of these control tissues individually to the TEF. These TEF mostly resembled esophagus based on the lowest number of differentially expressed genes (S1B in S1 File) between TEF and esophagus. The most differentially expressed genes (top 10) when comparing esophagus to TEF and when comparing trachea to TEF are depicted in Table 2, the 50 most differentially expressed genes are depicted in S1C in S1 File (esophagus vs. TEF) and S1D in S1 File (trachea vs TEF). We determined if sex was a biological variable and compared the transcriptomes of male (n = 13) and female (n = 8) TEF (FDR corrected, Foldchange >1.5). Apart from chromosome Y expressed genes (n = 10, S1E in S1 File) there were no differences.

### Transcriptome analysis: Pathway enrichment analysis

We studied the enrichment of pathways by the differentially expressed genes (n = 1045, at FDR p-value 0.01) when comparing the human TEF to esophageal and tracheal controls (S11 File). Pathways affected (p<0.05, Z-score of the direction change of the pathway at least +/- 1.5) are often related to cell adhesion and the extracellular matrix, the actin cytoskeleton, neuronal development and smooth muscle cell functioning (Table 3). In total, 28 genes were not

**Table 2. Top10 differential expressed genes.**

| Rank E vs TEF | Symbol | Fold change | TEF | E | T | L | Name | EntrezID |
|---|---|---|---|---|---|---|---|---|
| 1 | KCNMB1 | -16.888 | 1719.36 | 101.81 | 141.73 | 80.43 | potassium calcium-activated channel subfamily M regulatory beta subunit 1 | 3779 |
| 2 | SYNM | -19.903 | 5323.59 | 267.48 | 178.14 | 144.79 | synemin | 23336 |
| 3 | FAM83D | -10.988 | 698.96 | 63.61 | 91.31 | 81.83 | family with sequence similarity 83 member D | 81610 |
| 4 | CNN1 | -40.469 | 6108.81 | 150.95 | 233.83 | 124.38 | calponin 1 | 1264 |
| 5 | SYNPO2 | -30.756 | 3708.21 | 120.57 | 118.79 | 40.65 | synaptopodin 2 | 171024 |
| 6 | PLN | -22.613 | 1017.83 | 45.01 | 165.66 | 73.64 | phospholamban | 5350 |
| 7 | MBNL1-AS1 | -26.083 | 768.68 | 29.47 | 35.52 | 21.29 | MBNL1 antisense RNA 1 | 401093 |
| 8 | CSRP1 | -4.806 | 5547.58 | 1154.23 | 878.38 | 972.03 | cysteine and glycine rich protein 1 | 1465 |
| 9 | SMTN | -12.793 | 1616.07 | 126.32 | 157.95 | 160.03 | smoothelin | 6525 |
| 10 | LMOD1 | -16.01 | 1156.59 | 72.24 | 58.12 | 27.13 | leiomodin 1 | 25802 |
| 1 | ACTG2 | -9.879 | 12547.21 | 975.48 | 1270.12 | 604.38 | actin, gamma 2, smooth muscle, enteric | 72 |
| 2 | CNN1 | -26.125 | 6108.81 | 150.95 | 233.83 | 124.38 | calponin 1 | 1264 |
| 3 | ASB2 | -19.324 | 193.24 | 35.64 | 10 | 12.34 | ankyrin repeat and SOCS box containing 2 | 51676 |
| 4 | SYNM | -29.884 | 5323.59 | 267.48 | 178.14 | 144.79 | synemin | 23336 |
| 5 | DES | -51.004 | 3623.3 | 136.06 | 71.04 | 51.99 | desmin | 1674 |
| 6 | TPM2 | -8.798 | 8613.55 | 994.58 | 979.01 | 557.9 | tropomyosin 2 | 7169 |
| 7 | SYNPO2 | -31.217 | 3708.21 | 120.57 | 118.79 | 40.65 | synaptopodin 2 | 171024 |
| 8 | SLC26A7 | 21.765 | 13.06 | 283.76 | 284.25 | 54.38 | solute carrier family 26 member 7 | 115111 |
| 9 | KCNMB1 | -12.131 | 1719.36 | 101.81 | 141.73 | 80.43 | potassium calcium-activated channel subfamily M regulatory beta subunit 1 | 3779 |
| 10 | HACD1 | -13.187 | 751.91 | 181.48 | 57.02 | 83.98 | 3-hydroxyacyl-CoA dehydratase 1 | 9200 |

Depicted are the geometric measures of intensity (GMI) the expression signatures of the top 10 differentially expressed genes between Esophagus and TEF and the top 10 differentially expressed genes between Trachea and TEF. Genes are ranked on their pairwise parametric P-value, which were all below 0.0000001. The GMI intensity boxes are labeled in a color scale from orange (low) to blue (high). Statistical analysis was done using a multivariate permutation test with a maximum proportion of false discoveries of 0.01, a confidence level of 0.8 1000 permutations. Top 50 genes are depicted in the supplementary data.

expressed in the control tissues, but were expressed in more than half (n≥11) TEFs (S2A in S2 File). Vice versa, 40 genes were expressed in all controls, but lacked expression in more than half (n≥11) of the TEFs (S2B in S2 File, S13 File) Absence or presence of gene expression could hint at dysregulation of specific genes, pathways or processes. There were no significant biological processes, molecular functions or cellular component enriched in these two gene sets compared to the Homo sapiens reference set.

## Transcriptome analysis: Comparison to mouse foregut expression data

We evaluated if genes important for foregut development are differentially expressed between TEF and controls. For this we, used publicly available mouse gene expression data (GSE13040, GSE19873) [83, 84] at different time points (E8.25-E11.5) (S13 File). Indeed, 798 out of the 986 genes with a mouse orthologue gene were also differentially expressed in the mouse foregut across key mouse foregut developmental milestones (E8.5-E11.5) and could be of importance for proper foregut separation. Furthermore, several genes of which animal knockouts develop TEF were differentially expressed between TEF and trachea (Table 4) (MEOX2 downregulation) and between TEF and both trachea and esophagus (FOXF1 upregulation, SOX4 and DYNC2H1 downregulation).

**Table 3. Enriched pathways.**

| | Canonical Pathways | Z score | E vs TEF | Z score | T vs TEF | Remarks |
|---|---|---|---|---|---|---|
| 1 | Integrin Signaling | -3.000 | 6.488 | -3.000 | 6.488 | ECM, AC |
| 2 | HOTAIR Regulatory Pathway | 1.789 | 4.535 | 0.894 | 4.535 | D |
| 3 | Paxillin Signaling | -2.714 | 4.039 | -2.714 | 4.039 | ECM, AC |
| 4 | Superpathway of D-myo-inositol (1,4,5)-trisphosphate Metabolism | -1.633 | 3.386 | -1.633 | 3.386 | SMC? |
| 5 | TCA Cycle II (Eukaryotic) | -2.449 | 3.278 | -2.449 | 3.278 | E |
| 6 | D-myo-inositol (1,4,5)-trisphosphate Degradation | -2.236 | 3.157 | -2.236 | 3.157 | SMC? |
| 7 | Calcium Signaling | -2.138 | 3.015 | -1.604 | 3.015 | SMC |
| 8 | PTEN Signaling | 1.604 | 2.818 | 1.604 | 2.818 | D; AP |
| 9 | Actin Cytoskeleton Signaling | -1.698 | 2.717 | -1.698 | 2.717 | AC; SMC |
| 10 | ERK/MAPK Signaling | -1.886 | 2.579 | -1.414 | 2.579 | D |
| 11 | Regulation of Actin-based Motility by Rho | -2.111 | 2.504 | -2.111 | 2.504 | AC; SMC |
| 12 | Signaling by Rho Family GTPases | -1.698 | 2.489 | -1.698 | 2.489 | AC; SMC |
| 13 | Salvage Pathways of Pyrimidine Ribonucleotides | -1.508 | 2.398 | -1.508 | 2.398 | |
| 14 | Ephrin B Signaling | 1.890 | 2.331 | 1.890 | 2.331 | D |
| 15 | Cardiac Hypertrophy Signaling | -2.065 | 2.243 | -1.606 | 2.243 | |
| 16 | IGF-1 Signaling | -2.121 | 2.169 | -2.121 | 2.169 | D; e.g. activates 10 and 29 |
| 17 | BMP signaling pathway | -1.890 | 1.868 | -1.134 | 1.868 | D |
| 18 | Actin Nucleation by ARP-WASP Complex | -2.121 | 1.830 | -2.121 | 1.830 | AC; SMC |
| 19 | CDK5 Signaling | -0.333 | 1.643 | -1.667 | 1.643 | AC; N |
| 20 | Agrin Interactions at Neuromuscular Junction | -2.121 | 1.609 | -2.121 | 1.609 | D; (S)MC; N |
| 21 | Thrombin Signaling | -2.111 | 1.606 | -1.508 | 1.606 | |
| 22 | Glioma Signaling | -1.667 | 1.594 | -1.000 | 1.594 | |
| 23 | Netrin Signaling | -1.890 | 1.592 | -1.890 | 1.592 | N |
| 24 | Endocannabinoid Cancer Inhibition Pathway | -2.111 | 1.571 | -2.111 | 1.571 | |
| 25 | Gluconeogenesis I | -2.000 | 1.568 | -2.000 | 1.568 | E |
| 26 | Apelin Liver Signaling Pathway | 2.000 | 1.568 | 2.000 | 1.568 | |
| 27 | Neuregulin Signaling | -1.890 | 1.556 | -1.890 | 1.556 | N |
| 28 | B Cell Receptor Signaling | -1.732 | 1.416 | -1.155 | 1.416 | |
| 29 | PI3K/AKT Signaling | -1.508 | 1.311 | -0.905 | 1.311 | D; ECM |
| 30 | Regulation of eIF4 and p70S6K Signaling | -1.890 | 1.307 | -1.134 | 1.307 | |

Depicted are the canonical pathways at a significance level of 1.3 (-log(p<0.05)) and a minimum z-score of 1.5 in either directional change of the pathway(n = 30). Pathways are derived by uploading the most significant (FDR, p-value 0.01) differential expressed genes from both the pairwise analysis of TEF vs Esophagus and TEF vs Trachea. E; esophagus, T; trachea, Pathways with functions in ECM; extracellular matrix organization, AC; actin cytoskeleton, D; development, AP; anterior-posterior axis formation, SMC; smooth muscle cell development or functioning, E; energy metabolism, N; neuronal development or functioning.

## Transcriptome analysis: Disease genes and known biological pathways

We determined the expression signatures of syndromal EA genes (n = 114) [64, 85] to see if we could detect differentially expressed genes between tissue types directly related to known genetic actors. One gene was upregulated in all individual TEF samples compared to all other individual esophageal, tracheal and lung controls: the actin binding cytoskeletal protein filamin A (*FLNA*) (Table 4), whilst *FREM2*, *CDH7* and *EFNB2* are downregulated in most individual samples (and differ significantly on a group level). Next, we focused on targeted differential expression analysis of genes from the best described and currently known pathways in human and mouse foregut morphogenesis. This resulted in the identification of 50 genes that were differentially expressed in TEF compared to either lung, trachea and/or

**Table 4. (Candidate) disease genes.**

| Symbol | Parametric P-Value | FDR | Permutation p-value | TEF | E | T | L | | EntrezID | Pairwise significant |
|---|---|---|---|---|---|---|---|---|---|---|
| *FLNA* | < 1e-07 | < 1e-07 | < 1e-07 | 2475.66 | 342.01 | 373.05 | 425.47 | filamin A | 2316 | (E, TEF), (L, TEF), (T, TEF) |
| *FREM2* | < 1e-07 | < 1e-07 | 1.00E-04 | 27.35 | 201.78 | 56.96 | 580.9 | FRAS1 related extracellular matrix protein 2 | 341640 | (E, TEF), (E, L), (E, T), (L, TEF), (L, T) |
| *MEOX2* | < 1e-07 | < 1e-07 | < 1e-07 | 275.68 | 467.28 | 827.06 | 1579.52 | mesenchyme homeobox 2 | 4223 | (E, TEF), (E, L), (L, TEF), (T, TEF), (L, T) |
| *FOXF1* | < 1e-07 | < 1e-07 | < 1e-07 | 632.2 | 289.48 | 82.27 | 805.29 | forkhead box F1 | 2294 | (E, TEF), (E, L), (E, T), (T, TEF), (L, T) |
| *SOX4* | 2.00E-07 | 2.00E-06 | < 1e-07 | 30.98 | 125.13 | 164.47 | 190.33 | SRY-box 4 | 6659 | (E, TEF), (L, TEF), (T, TEF) |
| *DYNC2H1* | 4.00E-07 | 3.33E-06 | < 1e-07 | 29.98 | 64.02 | 83.64 | 136.1 | dynein cytoplasmic 2 heavy chain 1 | 79659 | (E, TEF), (E, L), (L, TEF), (T, TEF) |
| *ROBO2* | 7.00E-07 | 5.00E-06 | < 1e-07 | 32.07 | 90.24 | 83.42 | 244.19 | roundabout guidance receptor 2 | 6092 | (E, TEF), (E, L), (L, TEF), (T, TEF), (L, T) |
| *FOXC2* | 2.20E-06 | 1.38E-05 | 3.00E-04 | 57.92 | 68.94 | 188.99 | 48.69 | forkhead box C2 | 2303 | (E, T), (T, TEF), (L, T) |
| *TBX5* | 2.45E-05 | 0.000136 | 4.00E-04 | 228.89 | 120.15 | 55.34 | 789.89 | T-box 5 | 6910 | (E, L), (L, TEF), (T, TEF), (L, T) |
| *CHD7* | 9.62E-05 | 0.000481 | 1.00E-04 | 112.55 | 239.01 | 170.67 | 220.02 | chromodomain helicase DNA binding protein 7 | 55636 | (E, TEF), (L, TEF), (T, TEF) |
| *COL3A1* | 0.000221 | 0.001 | 0.0011 | 179.57 | 676.8 | 629.91 | 336.57 | collagen type III alpha 1 chain | 1281 | (E, TEF), (T, TEF) |
| *EFNB2* | 0.000296 | 0.00123 | 0.0015 | 160.33 | 201.78 | 277.52 | 534.77 | ephrin B2 | 1948 | (E, L), (L, TEF) |
| *FGFR2* | 0.000421 | 0.00162 | 0.0033 | 277.79 | 342.17 | 232.7 | 996.42 | fibroblast growth factor receptor 2 | 2263 | (E, L), (L, TEF), (L, T) |
| *HRAS* | 0.000613 | 0.00219 | 0.001 | 154.97 | 107.99 | 79.73 | 68.07 | HRas proto-oncogene, GTPase | 3265 | (L, TEF), (T, TEF) |
| *KIF3A* | 0.002052 | 0.00684 | 0.0016 | 54.51 | 77.26 | 93.07 | 97.43 | kinesin family member 3A | 11127 | (L, TEF), (T, TEF) |
| *SEMA3E* | 0.002864 | 0.00895 | 0.0062 | 52.25 | 34.47 | 14.88 | 113.35 | semaphorin 3E | 9723 | (E, L), (T, TEF), (L, T) |
| *DACT1* | 0.004969 | 0.0146 | 0.0074 | 159.64 | 180 | 390.57 | 308.14 | dishevelled binding antagonist of beta catenin 1 | 51339 | (L, TEF), (T, TEF) |
| *RARA* | 0.005746 | 0.016 | 0.0104 | 78.7 | 82.26 | 244.88 | 18.05 | retinoic acid receptor alpha | 5914 | (E, L), (L, TEF), (L, T) |
| *FOXC1* | 0.011795 | 0.0305 | 0.023 | 95.07 | 42.1 | 208.95 | 36.94 | forkhead box C1 | 2296 | (E, T), (L, TEF), (L, T) |
| *ITGA4* | 0.012194 | 0.0305 | 0.0196 | 21.71 | 24.9 | 48.67 | 56.07 | integrin subunit alpha 4 | 3676 | (L, TEF), (T, TEF) |
| *FOXP2* | 0.033228 | 0.0791 | 0.0447 | 360.41 | 158.45 | 188.3 | 308.76 | forkhead box P2 | 93986 | (E, TEF) |
| *NIPBL* | 0.044314 | 0.0963 | 0.0572 | 84.51 | 157.38 | 125.07 | 163.82 | NIPBL, cohesin loading factor | 25836 | (L, TEF) |
| *CC2D2A* | 0.046041 | 0.0963 | 0.0498 | 148.68 | 107.28 | 184.85 | 203.1 | coiled-coil and C2 domain containing 2A | 57545 | (E, L), (E, T) |
| *MYCN* | 0.04623 | 0.0963 | 0.0654 | 34.47 | 58.01 | 59.03 | 84.56 | MYCN proto-oncogene, bHLH transcription factor | 4613 | (L, TEF) |

Depicted are the geometric measures of intensity (GMI) the expression signatures of differentially expressed (candidate-) EA disease genes [64, 85]: (1) Esophagus, (2) TEF, (3) Lung and (4) Trachea. Pairwise significance is depicted in the last column. The GMI intensity boxes are labeled in a color scale from orange (low) to blue (high). For example: Highly upregulated in TEF is the expression of FLNA compared to all control tissue types and downregulated is the expression of MEOX2. Genes are ranked on their pairwise class comparison according to the random variance t-test analysis. The columns are sorted by the parametric P-value, the false discovery rate (FDR) and the univariate permutation p-value.

esophagus controls and include key factors as *PTCH1*, *BMP2*, R-SMADS, I-SMADS and *SMAD4* (S3 File).

### Transcriptome analysis: Expression of cell type specific genes

Using gene sets representative for smooth muscle, enteric neurons, epithelium and chondro-cytes we determined if these signatures were also present in TEF. For instance, during human

enteric nervous system development enteric neural crest cells migrate through the foregut (week 4) and arrive in the distal hindgut (week 7) [86]. These cells form the enteric nervous system are critical in the control of smooth muscle cell functioning and intestinal motility [87]. We compared the expression patterns of genes involved in smooth muscle cell functioning, genes crucial in neuronal functioning and markers for neuronal subtypes and enteric neurons and glia specifically between TEF and controls.

Many smooth muscle contraction genes are overexpressed in TEF compared to esophagus as well as trachea (*KCNMB1*, *LMOD1*, *SMTN*, *CNN1*, *MYL9*, *MYOCD*, *ACTG2* and *MYLK*). The overexpression of these genes is likely the result of the large contribution of intestinal SMC in TEF as the smooth muscle enteric form of gamma 2 (*ACTG2*) is strongly upregulated in trachea (Table 2, S1C, S1D in S1 File and S4 File). Moreover, many neuronal genes (e.g. *KCNMB1*, *KCND3*, *KCNMA1*, *CHRM3*, *VIP*), are overexpressed in TEF, indicative of the presence of neurons in TEF. However, genes of the enteric nervous system are either not differentially expressed, or mostly higher in trachea (S5 and S6 Files). The trachea has a pseudostratified ciliated columnar epithelium (marked by high *KRT8* and *KRT18* expression) and the esophagus has a stratified squamous epithelium signature (marked by high *KRT14*, *KRT5*, *KRT1* and *KRT10* expression). TEF has high KRT8 as well as high KRT14 and KRT5, indicating that both ciliated and stratified epithelium might be present (S7 File). The trachea has cartilage rings. We used the cartilage markers described in S8 File, but could not get a clear cartilage signature as genes were differentially expressed across TEF and control tissues.

## Histology

TEF material of 8 patients (not evaluated using micro-array) were available for immunohistochemical staining. As expected, the mucosa of the esophagus consisted of squamous epithelium, whereas the trachea displayed ciliated epithelial cells (Fig 1). Hematoxylin (HE) and eosin stained slides showed squamous epithelium in the TEFs (Fig 2). The muscular layer of the TEFs showed variable degrees of disorganization. No cartilage was found in the TEF samples.

## Immunohistochemistry

Expression levels of key marker proteins (NKX2-1, BCL-2, MKI-67, RAR-α, RAR-β, SOX-2, BMP2, BMP4, BMPR1A, BMPR1B, BMPR2, Noggin) of foregut development were determined with immuno-histochemical staining. We evaluated if the results of the immunostaining were representative of the results of the transcriptome profiling. All TEF samples were negative for NKX2-1 immunostaining, as were normal trachea and esophagus (Table 5). BMP2 staining was absent in TEF, but positive in preterm and term esophagus and trachea (Fig 3). BMP4 staining was negative in all samples tested. The BMP receptors BMPR1A, BMPR1B and BMPR2 were positive in all samples. There was some RAR-alpha positive staining in term esophagus and TEF (Fig 3). RAR-beta staining was positive in TEFs, especially in the basal epithelium. All control samples were positive, where the trachea stained weakly and the esophagus strongly positive. SOX2 nuclear expression was found throughout the epithelium of the TEFs, with very strong cytoplasmic staining on the luminal side and less strong, but still clearly present, granular staining on the basal side. The control samples all showed cytoplasmic as well as nuclear SOX2 staining. An overview of the results of all stainings can be found in Table 5.

## Discussion

In this study we used an unbiased whole transcriptome approach to characterize the TEF in detail in order to get more insight in their etiology. The esophagus and trachea are foregut-

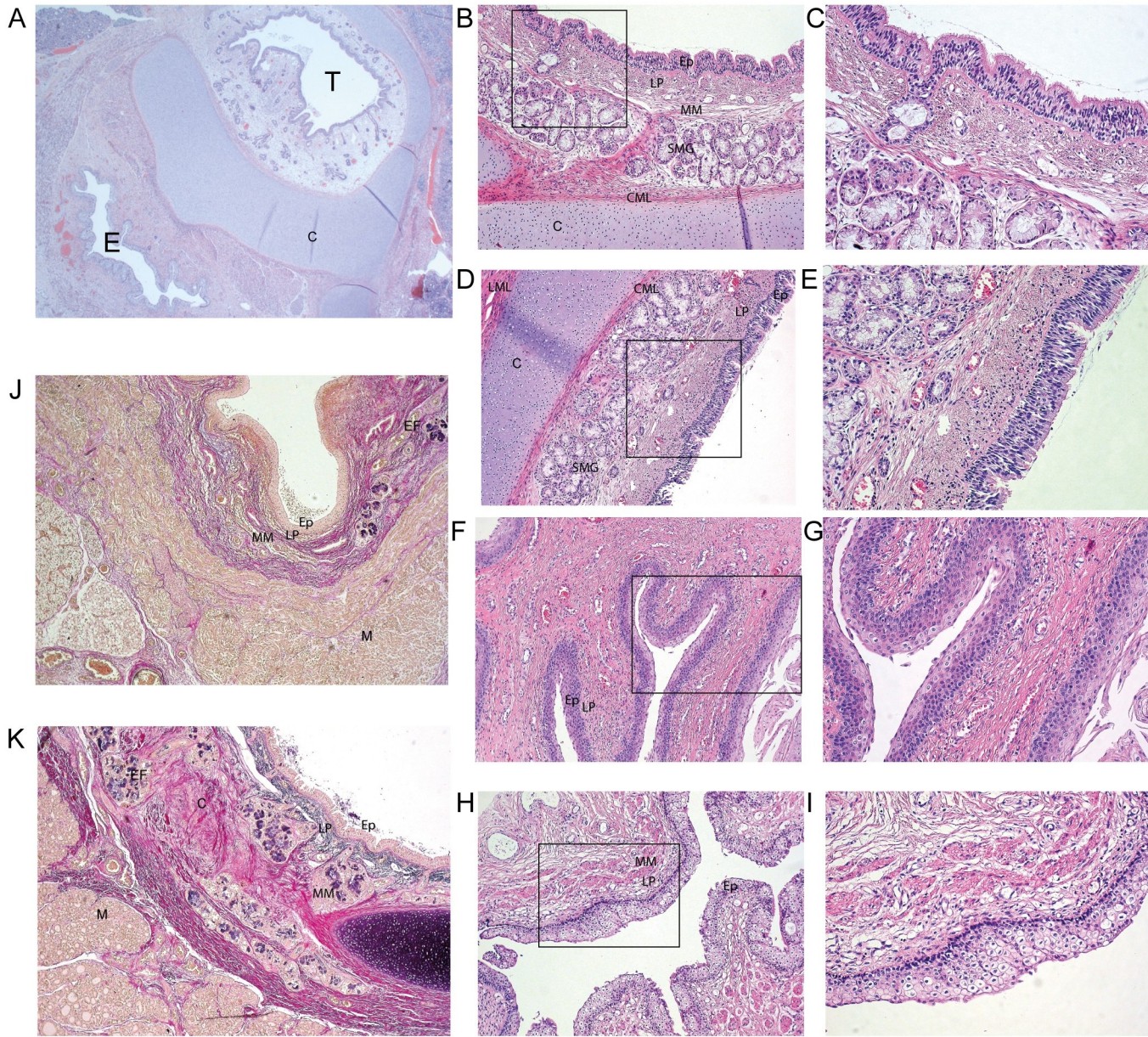

**Fig 1.** A-E. Hematoxylin and eosin (HE) stained sections of normal esophagus and trachea. A. Cross section of normal esophagus (E) and trachea (T) with surrounding cartilage (C) at 1.25 x magnification. B-C. Overview and detail of normal trachea with multilayered cylindrical epithelium (ep), underlying lamina propria (lp), muscular layer (mm) and seromucinous glands (smg); at the bottom of the image, the circular muscular layer (cml) and underlying cartilage can be seen. (at 10x (left) and 20x (right) magnification) C-D. Overview and detail of normal trachea covered with multilayered cylindrical epithelium (ep), with underlying glands, circular (cml) and longitudinal muscle layer (lml), both next to cartilage. (at 10x (left) and 20x (right) magnification) F-G and H-I. Normal esophagus covered by multilayered squamous epithelium (ep). (at 10x (left) and 20x (right) magnification) J and K. Elastic stain with elastic fibers in black (EF), collagen in pink (COL) and muscle (M) in yellow. In J an overview of normal esophagus is seen, with normal trachea in K. (at 4x magnification).

derived and during development there is a disturbance resulting in a faulty separation of these two structures. Unsupervised clustering analysis revealed that the most of the TEFs clustered separately from the controls and likely share more characteristics among each other on the level of gene expression than with these control tissues. TEF have large intestinal smooth muscle cell contribution, neuronal genes are expressed and there is likely ciliated and as well as

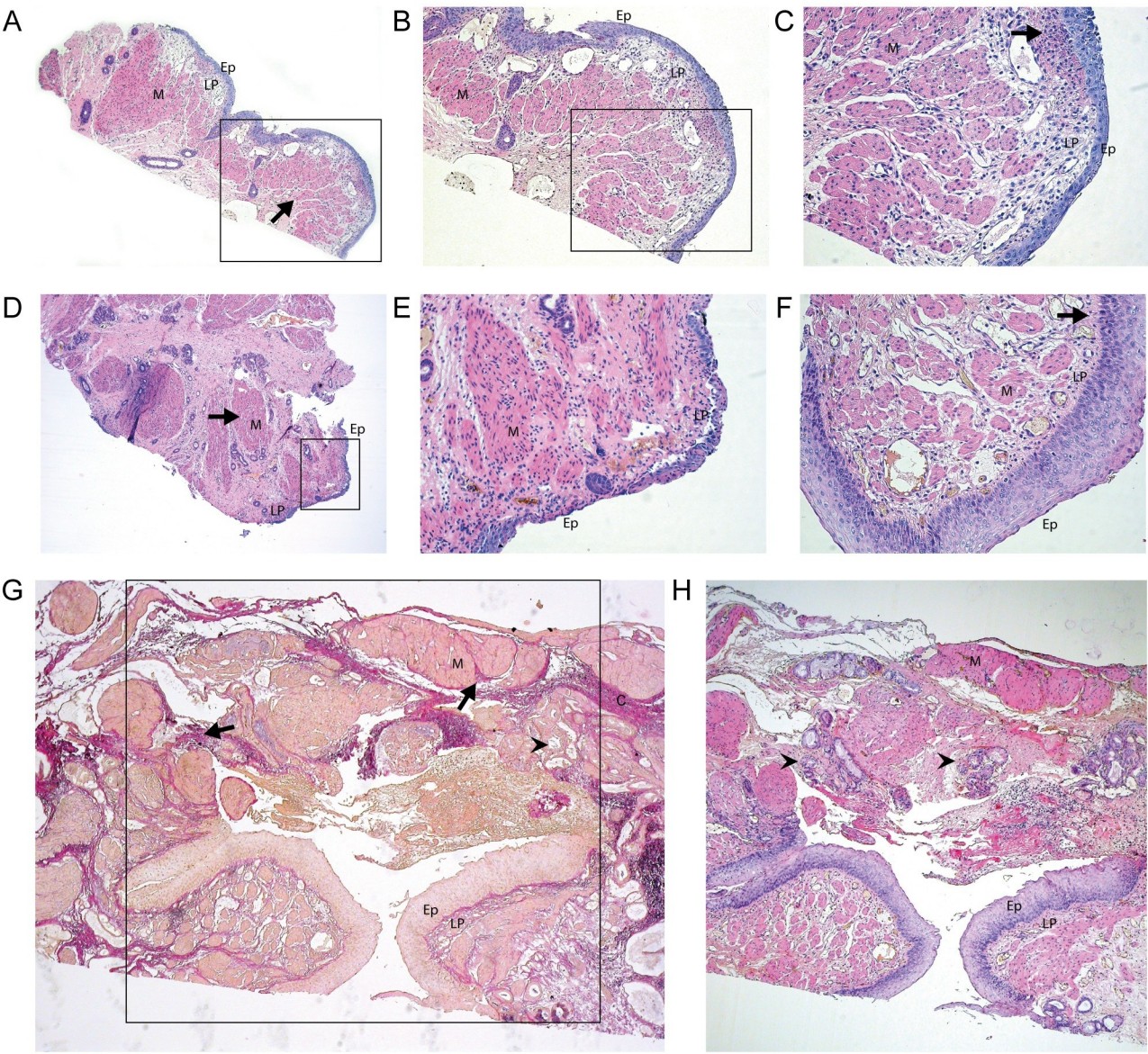

**Fig 2.** A-D. Overview and details of TEF. TEF walls covered by squamous epithelium (ep), with underlying lamina propria (lp) and muscular layer (m). The muscle layer appears irregular and fragmented in the overview of A-C and D, E and F. In the detail images a mild chronic inflammatory infiltrate can be appreciated. (at 4x, 10x and 40x magnification respectivly) G. Elastic and corresponding HE stain (H) of fistula covered with squamous epithelium, showing disorganized muscle bundles and glandular structures. (at 10x magnification).

stratified epithelium present. Transcription signatures and histological staining indicate that cell types normally present in esophagus are present in TEF. However, cell layers are often disorganized, which could be due to differences in exposure to signaling molecules. This combined would imply that the etiology of EA/TEF should not be sought in cell fate specification, but perhaps more in those biological processes involved in anterior-posterior or dorsal-ventral axis patterning or defects in signaling from the notochord or mesenchyme.

Two pathways known to be involved in tracheoesophageal development are affected. The first -he BMP pathway [49, 51]—is downregulated. Bone morphogenic proteins as Bmp4 and Bmp7 [49, 88] and their upstream regulator Noggin and Shh regulate dorsoventral patterning

**Table 5. Overview of the results of the immunohistochemical staining and differential expression analysis.**

|  | Term esophagus | Preterm esophagus | Term trachea | Preterm trachea | TEF |
|---|---|---|---|---|---|
| **NKX2-1** | - | - | - | - | - |
| **BCL-2** | - | +/- | + | + | - |
| *mRNA* | N/p | ++ | N/p | +++ | + |
| **(M)Ki-67** | + | N/p | + | N/p | + |
| **RAR-α** | +/- | - | - | - | +/- |
| *mRNA* | N/p | - | N/p | +++ | - |
| **RAR-β** | + | + | + | + | + |
| **SOX-2** | + | + | + | + | + |
| **BMP2** | + | + | + | + | - |
| *mRNA* | N/p | ++ | N/p | + | - |
| **BMP4** | - | - | - | - | - |
| **BMPR1A** | + | + | + | + | + |
| **BMPR1B** | + | + | + | + | + |
| **BMPR2** | + | + | + | + | + |
| **Noggin** | + | + | + | + | +[a] |
| **MMP-14** | +/- | N/p | +/- | N/p | + |
| *mRNA* | N/p | ++ | N/p | +++ | + |
| **MMP-2** | - epi.+ mes. | +/- | + | + | +/- epi.+ mes. |
| *mRNA* | N/p | ++ | N/p | +++ | + |

-: negative staining; +/-: variable results, some samples positive and some negative; + positive staining (mild); ++ positive staining (moderate); +++ positive staining (strong); N/p: not performed; epi: epithelium; mes: mesenchyme; a Mostly positive (see text for details) BCL2, RARα, and BMP2 were also differentially expressed between preterm esophagus, preterm trachea and TEF on mRNA level in the transcriptome samples. (M)Ki-67; marker of proliferation. BCL-2; apoptosis regulator. NKX2-1 staining were done on all patient samples, on all preterm samples and on four term trachea and esophagus samples. RAR-beta staining was done in all case samples, except for one TEF, in four term esophagus and term trachea samples and in all preterm samples. SOX2 staining was done on all case samples, except for one TEF. Furthermore, SOX2 was done on four term trachea and four term esophagus samples and on all preterm samples.

between endoderm and mesoderm and separation of the foregut into esophagus and the trachea [89]. We did not detect BMP2 protein expression and several genes of the TGF-β / BMP pathway (e.g. *BMP2*, R-SMADS (*SMAD1* & *SMAD5*) Co-SMAD (*SMAD4*) and I-SMADS (*SMAD6* & *SMAD7*) are downregulated. The second -he Ephrin B pathway—is upregulated in TEF. Ephrin B Signaling is upregulated. In contrast, *EFNB2* itself has the lowest expression when TEF is compared to esophagus, trachea and lung (Table 4). Interestingly, absent expression is related to tracheoesophgeal septation problems as *Efnb2* knockout mice develop TEF [90].

When comparing TEF to esophagus and trachea, we see that several interlinked pathways related to the actin cytoskeleton and adhesion to the extracellular matrix are downregulated in TEF (Table 3): integrins mediate cell adhesion to the ECM, link the ECM to the actin cytoskeleton, activate signal transduction pathways such as receptor tyrosine kinases [91, 92]. Coupling of the ECM to the actin cytoskeleton takes place through complexes of proteins such as integrin, vinculin, filamin and paxillin [93]. Paxillin is a scaffold enabling adhesion and growth factor molecules signaling between the plasma membrane and the actin cytoskeleton [94]. During development, Paxillin is involved in the development of mesoderm derived structures and has been shown to be a transducer of fibronectin signaling [95]. In Xenopus, syndecan-4 is required for fibronectin-1 extracellular matrix assembly, acts upstream of BMP and Wnt/ JNK signaling [96] and fibronectin 1 Xenopus have foregut defects. The PI3K/AKT Signaling pathway is also downregulated and is essential for endoderm formation [97]. It regulates the

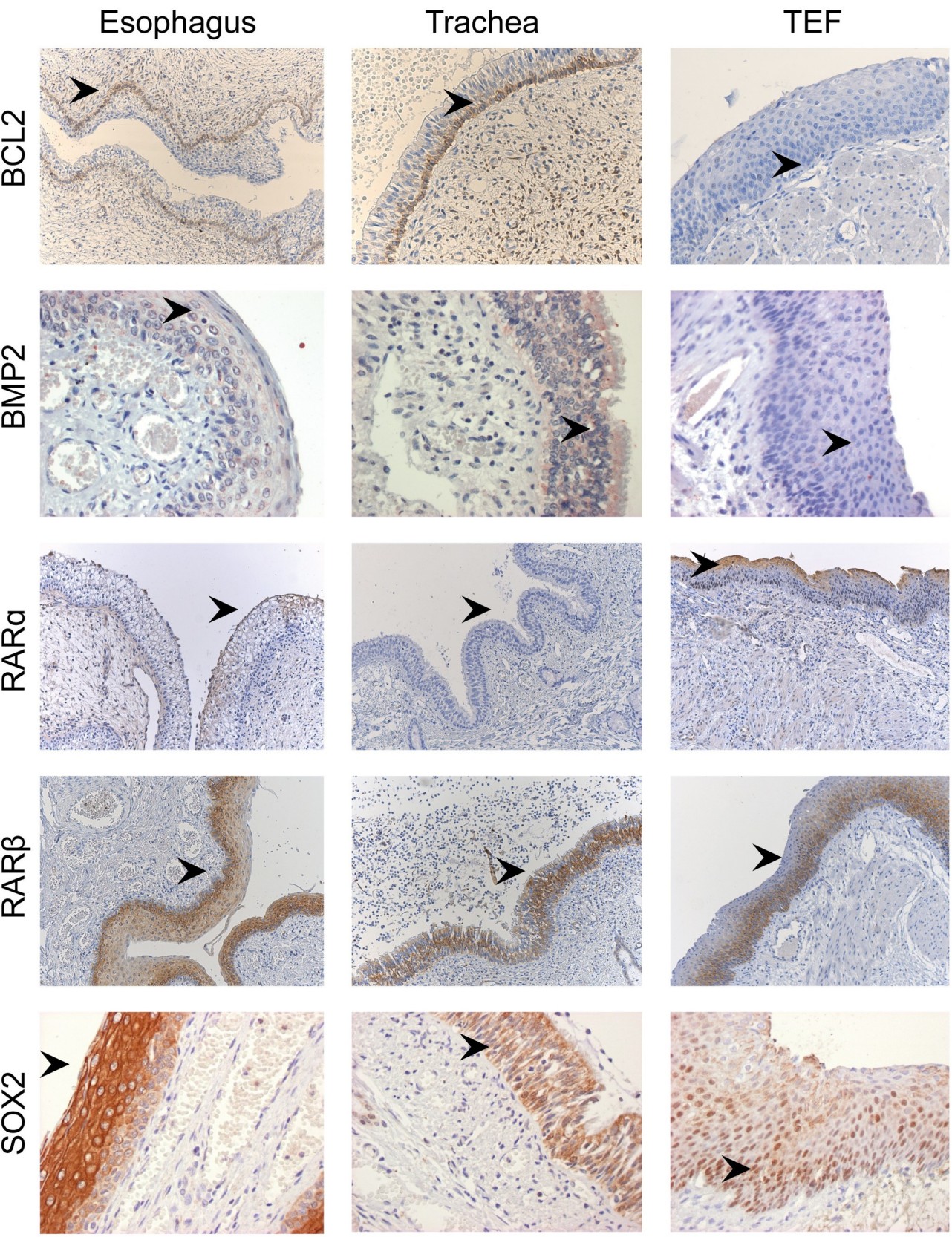

**Fig 3. Immunostaining of esophagus, trachea and TEF.** The first panel shows presence of bcl2 staining in the basal epithelial cells in both esophagus and trachea, but not in TEF. The second panel shows faint cytoplasmic BMP2 staining in esophagus and trachea, but not in TEF. The third panel shows cytoplasmic retinoic acid receptor alpha (RARα) staining in the upper part of the squamous epithelium of the esophagus and TEF, but not in the cylindrical epithelium of the trachea. In the fourth panel retinoic acid receptor beta (RARβ) shows similar cytoplasmic staining in the lower portion of the epithelium in all three structures. Interestingly, in the lower panel SOX2 shows cytoplasmic staining in esophagus and trachea, while there is evident nuclear labeling of epithelial cells in TEF. BLC2 esophagus and trachea at 10x magnification, TEF at 20x magnification. BMP2 and SOX2 at 40x magnification and RAR-α and RAR-β at 10x magnification.

levels of fibronectin in the foregut extracellular matrix. Without fibronectin and Integrin alpha 5 the foregut does not fold into a tube [98]. Connected to the extracellular matrix is the actin cytoskeleton. Actin cytoskeleton signaling, actin-based motility and other related pathways are also downregulated. CDK5 signaling is involved in the organization of the cytoskeleton and its contraction in both neurons and muscle [99–102].

Whilst many smooth muscle contraction related genes are upregulated in TEF, calcium signaling and pathways linking the extracellular matrix to the actin filament are downregulated. A process in which these pathways are entwined is myofibroblast directed fibrogenesis in which the actin stress fibers direct extracellular matrix remodelling [103, 104]. In response to TGFB1 fibroblast transform in smooth muscle like cells [105] with upregulation of *ACTA2, ACTG2 and* actin associated proteins are induced [106, 107]. BMPR1 activity is required for myoblast activation [108] and the fibrotic gene expression cascade [109] and mediates the abnormal proliferation of vascular smooth muscle cells seen in familial pulmonary arterial hypertension [110]. On transcriptome level, all evidence hints at a disruption of EMC-Cytoskeleton interaction. Immunohistochemistry of genes involved in this signaling cascade (BMPR1A and BMPR1B, MMP2 and MMP4) was not conclusive (Table 5, S10 File).

We performed immunohistochemical staining on some of the most crucial target molecules: e.g. NKX2-1, BMP2, BMP4, Retinoic Acid Receptor (RARα and β) and SOX2. As TEF is mostly derived from preterm infants, we compared TEF to term and preterm control esophagus and trachea. Several genes from the HH, WNT, BMP and retinoic acid signaling pathways remain differentially expressed (S3 File), most of them when comparing TEF to trachea and/or lung. This intriguing observation either indicate that these pathways *remain* disturbed after the original insult resulting in TEF, or indicate that these pathways are *no longer* essential in EA or lung development as TEF biopsies are taken after birth. High *NNKX2.1* and low *SOX2* characterizes future trachea and the opposite is true for future esophagus [111]. Noggin null mice form TEF which are lined with *Nkx2.1* expressing epithelial cells and indicate a respiratory origin [49]. However, *Nkx2.1* expression did not differ between any of the tissue types and TEF (Table 3). NKX2-1 was found by PCR and by immunohistochemistry in the epithelial tubules of the TEFs at term [74]. In rats with Adriamycin-induced EA/TEF, Nkx2.1 was also found by immunohistochemistry in the TEF throughout gestation, although its expression diminished later in pregnancy [78, 79]. The difference between these and our studies may reflect the heterogeneity of the TEFs or timing of developmental stage of controls and TEF.

Based on the theory that TEF originates from the tracheal bifurcation and grows down towards the stomach in a non-branching way, Crowley *et al* studied the expression pattern of *BMP2, 4* and *7* and *BMPR-IA, -IB* and *-II* in normal human lung, trachea and esophagus samples, as well as in samples of the proximal esophageal pouch and the TEF of nine patients with EA/TEF [75]. *BMP*-expression patterns in the proximal pouch were the same as in normal esophagus. The TEF tract showed a mixed pattern, with BMPs being absent (comparable to trachea) and also absence of the BMP-receptors, except for BMPR-II (comparable to esophagus), thereby confirming the hypothesis of an imbalance between ligands and receptors [75] Both mRNA and protein expression (see Table 5) of BMP2 and BMP4 are absent in TEF in

our study. There was no BMP4 protein expression nor is *BMP4* differentially expressed on mRNA level between preterm esophagus, preterm trachea or TEF. The BMPR2 receptor has protein expression throughout the tissues, including TEF. Thus in general, our findings are in line with these previous studies [75].

Two genes from the RA pathway (*ADH1B* and *RETSAT*) are involved in retinol metabolism, but are not crucial proteins involved in retinoic acid signaling. Similar patterns of immunohistochemical staining of these two retinoic acid receptors was observed in the lung, trachea and TEF specimens illustrative of the lack of difference in gene expression of genes involved in retinoic acid signaling. SOX2 is a transcription factor that, depending on its posttranslational status [112], should be predominantly present in the nucleus. Although the involvement of *SOX2* in the development of the foregut in general and EA/TEF in particular has been demonstrated in human and animal studies [37, 46, 49, 113–115], this involvement is not reflected the observed expression patterns of the different tissues in our study as we did not detect differences between TEFs and controls. Further studies, using more quantitative techniques may provide more detailed information on this.

We detect expression of most smooth muscle cell (SMC) contraction related genes (S 4) as well neuronal marker genes (S5 File), and enteric neuron and glia markers [116] (S6 File). There are strong indications that there are cells of the enteric nervous system and smooth muscle cells present in the TEF. As foregut separation occurs after ENCC migrate through the foregut, an ENS signature could be present in TEF. Indeed, the presence of SMC gene expression (e.g. *MYH11*, *MYL9*, *MYLK* and the intestinal specific *ACTG2*), potassium and calcium voltage gated channels (e.g. *KCNMB1*, *CACNA2D1*), neuronal subtype markers (e.g. *VIP*, *NPY*, *PENK*, *CARTP*), receptors (cholinergic, dopaminergic, GABBA) and ENS marker genes (*L1CAM*, *TUBB3*, *UCHL1*, *PRPH*, *ELAV4*, neurofilament and peripherin) further strengthen the evidence for a more intestinal program.

Using micro-array-, we can only determine the relative expression within a mix of patient cells. Further experiments in TEF and esophageal biopsies would benefit from a single cell approach, as this would allow for a detailed characterization and quantification of cell types. For instance, we cannot exclude tracheal SMC and neuronal contributions due to our experimental setup. Using single cell transcriptomics-based approaches we could have determined if *ACTG2* –negative SMC were present. Kishimoto et al. show that smooth muscle cell precursor polarization is the starting point for tracheal tube elongation [117]. The neuronal subtypes of trachea and esophagus mostly overlap, although there are esophageal specific neuronal cells [118]. Interestingly, most smooth muscle cell genes are strongly overexpressed compared to all tissue types hinting at a much larger SMC composition of the TEF biopsies and/or myofibroblast activation.

Several keratins are upregulated in TEF compared to all control tissues. *KRT8* and *KRT18* are markers for columnar epithelial markers, whilst *KRT14* and *KRT5* are expressed in the basal layer of stratified squamous epithelial cells, whilst *KRT1* and *KRT10* are expressed in the suprabasal layers [119]. Other markers for early epithelial differentiation include *Itgb4*, *Itgb6* and *Nt5e* (*Cd73*) [120]. During development there is a transition from columnar to pseudostratified epithelium. This transition is likely controlled by *TP63* [121]. Although we could not detect differential expression of *SOX2*, the upregulation of *NT5E* and *ITGB4* hint at the presence of early transition from basal to suprabasal cells. The esophageal epithelium-specific keratins (*KRT4* and *KRT13*) [122] are highly upregulated compared to the control tissues. Interestingly, we can measure upregulation of basal layer markers *KRT14* and *KRT5* as well as a comparable expression level of *KRT8* in esophagus and TEF. During that time there are also cells expressing both *KRT8* as well as *KRT14* [119]. We could not detect a clear cartilage specific signature in TEF.

This study characterizes the transcriptome of TEF and their histological composition.

We determined the relative gene expression of a mix of cells present in TEF and compared this to preterm esophagus, lung and trachea The number of control samples is low compared to the number of TEF. Including more control samples would allow for a more robust differential expression analysis. Furthermore, TEF are not naturally occurring tissue structures and it is not certain that the expression levels seen in this postnatal "end state" of development are representative of early development. Although all major cell types seem present, it is not certain that these cells would function normally. Future experiments using single cell sequencing would allow for a cell type specific comparison.

## Conclusions

Tracheoesophageal fistulas are fibrous tubular structures with large contributions of intestinal smooth muscle cells, mostly resembling the esophagus. TEF tissue layers are often structurally disorganized. We could not detect tracheobronchial remnants neither based on expression profiles nor on histological staining. The BMP-signaling pathway, actin cytoskeleton and extracellular matrix pathways are downregulated compared to esophagus and trachea. Pathways related to myofibroblast activated fibrosis are enriched. Additional experiments are required to determine if upregulation of genes involved in the actin cytoskeleton and smooth muscle cell functioning are related to the disorganized structure of the TEF, myoblast activated fibrosis or abnormal the functioning of these cell types [76, 123]. Furthermore, it is important to examine if these processes are disturbed throughout the esophagus and continue to affect neuromuscular functioning with disturbed esophageal motility as a consequence.

## Supporting information

**S1 File. Differential expression analysis, clustering and sex differences.**
(PDF)

**S2 File. TEF defining genes.**
(PDF)

**S3 File. DEGs in retinol metabolism, WNT-, TGFB- and hedgehog signaling.**
(PDF)

**S4 File. Smooth muscle contraction gene expression.**
(PDF)

**S5 File. Neuronal marker genes.**
(PDF)

**S6 File. DEG ENS markers.**
(PDF)

**S7 File. DEG epithelial markers.**
(PDF)

**S8 File. DEG cartilage markers.**
(PDF)

**S9 File. Details of antibodies used for immunohistochemistry.**
(PDF)

**S10 File. Immunostainings MMP2, MMP14 and BMPR1A.**
(PDF)

**S11 File. Causal network analysis using IPA<sup>©</sup>.**
(PDF)

**S12 File. TEF specific expression pattern analysis.**
(PDF)

**S13 File. Overlap with mouse developmental transcriptome.**
(PDF)

**S1 Graphical abstract. Transcriptome study and histochemistry of tracheoesophageal fistula.**
(TIF)

## Acknowledgments

The authors would like to thank the pediatric surgeons of the Erasmus MC–Sophia Children's Hospital, Monique Oomen of the Erasmus MC tissue bank, Tom de Vries-Lentsch for making the illustrations and Prof. Dr. Münnever Hösgör of the Dr. Behcet Uz Children's Hospital (Izmir, Turkey) for providing TEF material for (immune-) histological staining.

## Author Contributions

**Conceptualization:** Erwin Brosens, Janine F. Felix, Dick Tibboel, Robbert J. Rottier.

**Data curation:** Erwin Brosens.

**Formal analysis:** Erwin Brosens, Anne Boerema-de Munck, Elisabeth M. Lodder, Sigrid Swagemakers.

**Funding acquisition:** Erwin Brosens, Janine F. Felix, Anne Boerema-de Munck, Elisabeth M. de Jong, Annelies de Klein, Dick Tibboel.

**Investigation:** Anne Boerema-de Munck, Sigrid Swagemakers, Ronald R. de Krijger.

**Methodology:** Anne Boerema-de Munck, Elisabeth M. Lodder.

**Project administration:** Anne Boerema-de Munck, Elisabeth M. de Jong, Marjon Buscop-van Kempen.

**Resources:** Janine F. Felix, Anne Boerema-de Munck, Elisabeth M. de Jong, Rene M. H. Wijnen.

**Software:** Erwin Brosens, Sigrid Swagemakers, Wilfred F. J. van IJcken.

**Supervision:** Dick Tibboel, Robbert J. Rottier.

**Validation:** Marjon Buscop-van Kempen.

**Writing – original draft:** Erwin Brosens, Janine F. Felix, Robbert J. Rottier.

**Writing – review & editing:** Erwin Brosens, Janine F. Felix, Ronald R. de Krijger, Rene M. H. Wijnen, Wilfred F. J. van IJcken, Peter van der Spek, Annelies de Klein, Dick Tibboel, Robbert J. Rottier.

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
