## [Decision Letter · Decision Letter 0]

27 Aug 2020

PONE-D-20-17082

Histological, immunohistochemical and transcriptomic characterization of human tracheoesophageal fistulas

PLOS ONE

Dear Dr. Brosens,

Thank you for submitting your manuscript to PLOS ONE. After careful consideration, we feel that it has merit but does not fully meet PLOS ONE’s publication criteria as it currently stands. Therefore, we invite you to submit a revised version of the manuscript that addresses the points raised during the review process.

The reviewers identified several errors in the text and missing technical details that need to be added. If possible, one reviewer requests that gender be considered as a relevant variable.

We look forward to receiving your revised manuscript.

Kind regards,

David D. Roberts

Academic Editor

PLOS ONE

Journal Requirements:

2. Thank you for stating in the text of your manuscript that parental informed consent was obtained. Please clarify whether this was written or verbal consent. If verbal, please discuss how consent was witnessed and documented. Please also add all of this information to your ethics statement in the online submission form.

3. Please add the catalog numbers of the antibodies used in this study to table S10.

Reviewers' comments:

Reviewer's Responses to Questions

**Comments to the Author**

1. Is the manuscript technically sound, and do the data support the conclusions?

Reviewer #1: Yes

Reviewer #2: Partly

2. Has the statistical analysis been performed appropriately and rigorously? 

Reviewer #1: Yes

Reviewer #2: Yes

3. Have the authors made all data underlying the findings in their manuscript fully available?

Reviewer #1: Yes

Reviewer #2: Yes

4. Is the manuscript presented in an intelligible fashion and written in standard English?

Reviewer #1: Yes

Reviewer #2: Yes

5. Review Comments to the Author

Reviewer #1: This manuscript aims to define a transcriptional signature of tracheoesophageal fistulas in humans. Secondarily, they provide histological and immunohistochemical data for further characterization. In general, the manuscript is straight forward and a characterization/description paper. However, it does add new data to the field and will serve as a basis for further investigations and characterization. Below are comments that require attention.

Abstract

Nice abstract

Introduction

Line 67, page 3, please define “CNVs”

Materials and Methods

Please list the Institutional Review approval number for the human studies.

Please list inclusion and exclusion criteria for humans

Please also provide the demographics of the patient population

Please describe how RNA integrity was determined and the criteria using the RNA in downstream RNA-sequencing.

Please provide the sequencing depth and number of reads.

Please provide information about statistical analyses.

As a general note, there are details about methods provided in the supplemental material – it would be beneficial to include notes in the main manuscript pointing readers to the supplemental material for more details about experimental procedures.

Results

Figures 1-3. Arrows pointing to the salient features of interest would benefit the reader.

Page 13, line 223, a parenthesis is missing after KCNMB1

Table 4, please provide a reference for what the “+” means (for example how much more expression does +++ represent compared to +

In table S2b, there was no obvious data presented in the row “TEF vs Esophagus and Trachea”

For tables S2c-S3b, please indicate whether the p values were corrected using FDR.

For tables S2c-S3b, please indicate the comparison made in the fold change column (TEF vs ?)

For tables S2c-S3b, please also provide the units for columns TEF, E, T, L

In tables S4-S5, some number are italicized, is there a reason for this and/or does this represent something the reader should pay attention to?

Discussion

There are no mentions or discussion about sex as a biological variable or whether sex differences were probed for. The authors should consider adding a statement or two related to this.

In addition, the number of control samples is relatively small compared to the experimental population. The authors should include this as a limitation of their work and acknowledge that a larger control population might reveal more rigorous data/results.

The discussion would benefit from a final summary paragraph and key take home message and potential future directions.

Reviewer #2: Histological, immunohistochemical and transcriptomic characterization of human tracheoesophageal fistulas

Interesting, well-written, straightforward descriptive paper looking primarily at gene expression using transcriptomics, followed by some eval of tissue organization, and immunohistochemistry of key protein patterns in trachea-esophageal fistula samples vs esophageal, tracheal and lung control samples. Some issues, mostly minor addressed below.

Abstract: Not sure what last sentence means “This combined implies that EA/TEF etiology should not be sought in cell fate specification”.

Methods: Patient characteristics: Where did control tissues come from? This information along with characteristics of patients from which 21 TEF samples obtained is well-described in supplementary methods. Should this information be in the main text?

Results:

Overall, may want to elaborate results a bit more so that discussion is better supported.

Line 221: in this line and throughout the text, I am not clear what is meant by intestinal SMC in relation to TEFs or the esophagus; ACTG2 is most differentially expressed in T vs TEF – is this correct (Table 1)? Is ACTA2 present and in which tissue samples?

line 223 (KCNMB1 needs another parentheses

Line 263: verbiage – “both” needs to be removed; “RAR-beta staining was positive both in TEFs, especially in the basal epithelium.”

Discussion:

Not sure section headings in the discussion are helpful. Some editing needed e.g. Line 336 In response to TGFB1 fibroblast transform in smooth muscle like cells [104] with high ACTA2 (or the intestinal ACTG2?) in which actin associated proteins are induced[105]; also Minor editing: “There was no BMP4 protein expression nor is differentially expressed on mRNA level between preterm esophagus, preterm trachea or TEF.”

May want to address limitations of study – whole tissue was submitted; wasn’t single cell sequencing (briefly mentioned) which may limit interpretation of your findings.

6. PLOS authors have the option to publish the peer review history of their article (what does this mean?). If published, this will include your full peer review and any attached files.

Reviewer #1: No

Reviewer #2: No

---

## [Author Response · Author response to Decision Letter 0]

16 Oct 2020

Response to reviewer 1

We appreciate the detailed sugestions for improvement constructive comments fom the reviewers.

Introduction

• Line 67, page 3, please define “CNVs”

We added the definition of CNV to the sentence:

“Approximately 10% of patients with syndromal EA/TEF have chromosomal anomalies, mostly trisomies [1, 24, 25], deleterious Copy Number Variations (CNVs) [26-28] or a monogenetic syndrome [29-40].

Materials and Methods

• Please list the Institutional Review approval number for the human studies. 

We added the sentence (line 108-110): 

“Written (parental) consent was obtained. This study has been approved by the Erasmus University Medical Center’s local ethics board (protocol no.193.948/2000/159, addendum Nos. 1 and 2.)”

• Please list inclusion and exclusion criteria for humans

We included all samples born in our hosiptal (Erasmus MC) for the transriptome study of which parents gave informed consent and the operating surgeon considered it safe and technicaly feasible to remove tissue from the fistula. 

We added the section (line 110-114) : 

“After parental informed consent was obtained, tissue samples of the TEF of children with EA with a distal TEF were taken during primary operative repair of the EA/TEF. The operating surgeon, who had no involvement in the study, determined the safety and technical feasibility of removing the tissue.”

• Please also provide the demographics of the patient population

Patient characteristics were described in the supplementary data (S1). We agree a more detailed description would beneift the reader. Additionaly, reviewer 2 also suggested to move the patient characteristics and detailed descriptions to the main text. (line 115 – 142 and table 1)

• Please describe how RNA integrity was determined and the criteria using the RNA in downstream RNA-sequencing.

We added the sentence (line 153-155) 

“RNA quality was evaluated by inspecting ribosomal 28S and 18S peaks and using the bioanalyser (RNA integrity number (RIN) values above 8.0) Samples with low RNA quality were excluded from the transcriptome study. All samples were snap frozen in liquid nitrogen and stored at –80 °C until further processing)” 

• Please provide the sequencing depth and number of reads.

Transcriptome analyis was done using microarray, not RNA sequencing. Quality control steps are moved from the supplementary methods tot he main text. We added the sections: “RNA isolation and quality control” , and “Data processing and normalization” and “Class comparison of tissues types” to the main text (method section) to clarify these steps in more detail. 

• Please provide information about statistical analyses.

We included the statistical analysis in the main text (methods: class comparison of tissue types) as well as the supplementary methods (M1, M2 and M3)

• As a general note, there are details about methods provided in the supplemental material – it would be beneficial to include notes in the main manuscript pointing readers to the supplemental material for more details about experimental procedures.

In addition to moving parts of the supplementary methods tot he main texst, we refffered to supplementary method M1, M2 and M3 in the result section of the main text.

Results

• Figures 1-3. Arrows pointing to the salient features of interest would benefit the reader.

We have done so accordingly and described the figures in more detail in the figure legends. We noticed a numbering mistake in figure 1 description and correted this (the upper panel pictures are trachea, not esophagus)

• Page 13, line 223, a parenthesis is missing after KCNMB1

There was indeed something missing from the sentence. We corrected the sentence to:

“(e.g. KCNMB1, KCND3, KCNMA1, CHRM3, VIP)”

• Table 4, please provide a reference for what the “+” means (for example how much more expression does +++ represent compared to +

Immunohistochemistry was evaluated by two individuals, Indeed this evaluation is subjective and we have chosento evaluate the stainings from none (absent expression) to strong expression.

We added an description in the figure legend: . -: negative staining; +/-: variable results, some samples positive and some negative; + positive staining (mild); ++ positive staining (moderate); +++ positive staining (strong); N/p: not performed;

• In table S2b, there was no obvious data presented in the row “TEF vs Esophagus and Trachea”

Values were missing from the table, we added them (1397↑, 1259↓)

• For tables S2c-S3b, please indicate whether the p values were corrected using FDR.

Yes we corrected using FDR, but the depicted p-pvalues are from the parametric test. We added the FDR corrected values to the table. 

• For tables S2c-S3b, please indicate the comparison made in the fold change column (TEF vs ?)

Foldchange for each table was as described in the title. We added the description tot the table legend for clarity.

• For tables S2c-S3b, please also provide the units for columns TEF, E, T, L

We added the description to the table legend

• In tables S4-S5, some number are italicized, is there a reason for this and/or does this represent something the reader should pay attention to?

No there was no reason for this, we corrected this in the tables.

Discussion

• There are no mentions or discussion about sex as a biological variable or whether sex differences were probed for. The authors should consider adding a statement or two related to this.

We compared male and female TEFs and added the follow sentence the tekst (line 3-8):

“We determined if sex was a biological variable (S12) and compared the transcriptomes of male (n=13) and female (n=8) TEF (FDR corrected, Foldchange >1.5). Apart from chromosome Y expressed genes (n=10, S2e) there were no differences”.

• In addition, the number of control samples is relatively small compared to the experimental population. The authors should include this as a limitation of their work and acknowledge that a larger control population might reveal more rigorous data/results.

We added a parapgraph before the conclusion section (line 527-536) :

“This study characterizes the transcriptome of TEF and their histological composition. 

“We determined the relative gene expression of a mix of cells present in TEF and compared this to preterm esophagus, lung and trachea The number of control samples is low compared to the number of TEF. Including more control samples would allow for a more robust differential expression analysis. Furthermore, TEF are not naturally occurring tissue structures and it is not certain that the expression levels seen in this postnatal “end state” of development are representative of early development. Future experiments using single cell sequencing would allow for a cell type specific comparison.” 

• The discussion would benefit from a final summary paragraph and key take home message and potential future directions.

Future directions are briefly mentioned in (line 535-536):

Future experiments using single cell sequencing would allow for a cell type specific comparison.

And in line 538-550:

We rephrased the conclusion section to: 

“Tracheoesophageal fistulas are fibrous tubular structures with large contributions of intestinal smooth muscle cells, mostly resembling the esophagus. TEF tissue layers are often structurally disorganized. We could not detect tracheobronchial remnants neither based on expression profiles nor on histological staining. The BMP-signaling pathway, actin cytoskeleton and extracellular matrix pathways are downregulated compared to esophagus and trachea. Pathways related to myofibroblast activated fibrosis are enriched. Additional experiments are required to determine if upregulation of genes involved in the actin cytoskeleton and smooth muscle cell functioning are related to the disorganized structure of the TEF, myoblast activated fibrosis or abnormal the functioning of these cell types [76, 122]. Furthermore, it is important to examine if these processes are disturbed throughout the esophagus and continue to affect neuromuscular functioning with disturbed esophageal motility as a consequence”.

 

Response to reviewer 2

We appreciate the detailed sugestions for improvement constructive comments fom the reviewers.

Abstract: 

• Not sure what last sentence means “This combined implies that EA/TEF etiology should not be sought in cell fate specification”.

With this, we imply that all major cell types present in esophagus are present in TEF. Absence of a specific cell type as a result of defects in cell differentiation / maturation does not seem to be a likely cause for EA/TEF. We rephrased the sentence to:

“All major cell types present in esophagus are present - albeit often structurally disorganized - in TEF, indicating that its etiology should not be sought in cell fate specification”.

Methods: 

• Patient characteristics: Where did control tissues come from? This information along with characteristics of patients from which 21 TEF samples obtained is well-described in supplementary methods. Should this information be in the main text?

We agree this would benefit the reader and movd the table and information information to the main text

Results:

Overall, may want to elaborate results a bit more so that discussion is better supported.

We added:

“Absence or presence of gene expression could hint at dysregulation of specific genes, pathways or processes (line 245-246)”

We rephrased the mouse transcriptome paragraph to:

“We evaluated if genes important for foregut development are differentially expressed between TEF and controls. For this we, used publicly available mouse gene expression data (GSE13040, GSE19873) (83, 84) at different time points (E8.25-E11.5) (M3). Indeed, 798 out of the 986 genes with a mouse orthologue gene were also differentially expressed in the mouse foregut across key mouse foregut developmental milestones (E8.5-E11.5) and could be of importance for proper foregut separation. Furthermore, several genes of which animal knockouts develop TEF were differentially expressed between TEF and trachea (Table 4) (MEOX2 downregulation) and between TEF and both trachea and esophagus (FOXF1 upregulation, SOX4 and DYNC2H1 downregulation).”

• Line 221: in this line and throughout the text, I am not clear what is meant by intestinal SMC in relation to TEFs or the esophagus; ACTG2 is most differentially expressed in T vs TEF – is this correct (Table 1)? Is ACTA2 present and in which tissue samples?

ACTG2 (top 36 TEF vs Esophagus (S1c) and top 1 when comparing trachea to TEF S1d) is a smooth muscle actin subtype mainly expressed in the gastrointestinal tract, arteries and bladder (https://gtexportal.org/home/gene/ACTG2), ACTA2 is more broadly expressed (https://gtexportal.org/home/gene/ACTA2) As this analysis is done with micro-array, we can only determine relative expression (higher/lower) and speculate about the presence of gene expression(absent/present). ACTA2 is higher expressed in Esophagus compared to TEF (FDR 0.0383, FC 3.57), but not differentially expressed between TEF and trachea (hinting at the same expression level of Trachea and Esophagus)

• line 223 (KCNMB1 needs another parentheses

There was indeed something missing from the sentence. We corrected the sentence to:

“(e.g. KCNMB1, KCND3, KCNMA1, CHRM3, VIP)”

• Line 263: verbiage – “both” needs to be removed; “RAR-beta staining was positive both in TEFs, especially in the basal epithelium.”

We removed “both” from the sentence

Discussion:

• Not sure section headings in the discussion are helpful. 

We removed the section headings from the discussion

• Some editing needed e.g. Line 336 In response to TGFB1 fibroblast transform in smooth muscle like cells [104] with high ACTA2 (or the intestinal ACTG2?) in which actin associated proteins are induced[105]; 

We rephrased the senetence and added a reference for ACTG2 upregulation

In response to TGFB1 fibroblast transform in smooth muscle like cells [105] with upregulation of ACTA2, ACTG2 and actin associated proteins are induced[106].

• Minor editing: “There was no BMP4 protein expression nor is differentially expressed on mRNA level between preterm esophagus, preterm trachea or TEF.”

We rephrased the sentence to:

“There was no BMP4 protein expression nor is BMP4 differentially expressed on mRNA level between preterm esophagus, preterm trachea or TEF.”

May want to address limitations of study – whole tissue was submitted; wasn’t single cell sequencing (briefly mentioned) which may limit interpretation of your findings.

We added the following to the tekst (line 501-507)

“Using micro-array-, we can only determine the relative expression within a mix of patient cells. Further experiments in TEF and esophageal biopsies would benefit from a single cell approach, as this would allow for a detailed characterization and quantification of cell types. For instance, we cannot exclude tracheal SMC and neuronal contributions due to our experimental setup. Using single cell transcriptomics-based approaches we could have determined if ACTG2 – negative SMC were present.”

---

## [Decision Letter · Decision Letter 1]

28 Oct 2020

Histological, immunohistochemical and transcriptomic characterization of human tracheoesophageal fistulas

PONE-D-20-17082R1

Dear Dr. Brosens,

We’re pleased to inform you that your manuscript has been judged scientifically suitable for publication and will be formally accepted for publication once it meets all outstanding technical requirements.

Kind regards,

David D. Roberts

Academic Editor

PLOS ONE

Additional Editor Comments (optional):

Please correct the remaining typographical error identified by one reviewer.

Reviewers' comments:

Reviewer's Responses to Questions

**Comments to the Author**

1. If the authors have adequately addressed your comments raised in a previous round of review and you feel that this manuscript is now acceptable for publication, you may indicate that here to bypass the “Comments to the Author” section, enter your conflict of interest statement in the “Confidential to Editor” section, and submit your "Accept" recommendation.

Reviewer #1: All comments have been addressed

Reviewer #2: All comments have been addressed

2. Is the manuscript technically sound, and do the data support the conclusions?

Reviewer #1: Yes

Reviewer #2: Yes

3. Has the statistical analysis been performed appropriately and rigorously? 

Reviewer #1: Yes

Reviewer #2: Yes

4. Have the authors made all data underlying the findings in their manuscript fully available?

Reviewer #1: Yes

Reviewer #2: Yes

5. Is the manuscript presented in an intelligible fashion and written in standard English?

Reviewer #1: Yes

Reviewer #2: Yes

6. Review Comments to the Author

Reviewer #1: The authors have addressed my concerns and the manuscript is now strengthened and improved. The only thing I noted on the revision is the typo or undefined term "-he" found in the discussion.

Reviewer #2: The authors have addressed all my comments.

Manuscript is technically sound and presented in an intelligible fashion.

7. PLOS authors have the option to publish the peer review history of their article (what does this mean?). If published, this will include your full peer review and any attached files.

Reviewer #1: No

Reviewer #2: No

---

## [Editor Report · Acceptance letter]

4 Nov 2020

PONE-D-20-17082R1 

Histological, immunohistochemical and transcriptomic characterization of human tracheoesophageal fistulas 

Dear Dr. Brosens:

I'm pleased to inform you that your manuscript has been deemed suitable for publication in PLOS ONE. Congratulations! Your manuscript is now with our production department. 

Kind regards, 

on behalf of

Dr. David D. Roberts 

Academic Editor

PLOS ONE